# ALS: Attentive Long-Short-Range Message Passing

## Abstract

The graph attention (GAT) mechanism has been instrumental in enabling nodes to aggregate information from their neighbours based on relevancies, significantly enhancing the adaptiveness of graph neural networks across various graph representation learning tasks. Recent research has sought to further leverage the power of GAT for tasks that require capturing long-range data dependencies. However, the conventional stacking of GAT layers leads to excessive memory footprint, computation overhead, and the issue of over-smoothing. To address these challenges, this study proposes Attentive Long-Short-range message passing (ALS), which integrates personalized PageRank to mitigate the over-smoothing problem in long-range message passing and leverages GAT to capture complex data dependencies. Compared with the naive $L$-step message passing, which has a space complexity of $O(L)$ for its optimization, ALS employs implicit differentiation to achieve $O(1)$ memory footprint and three acceleration techniques to reduce up to 89.51% computation time. Extensive experiments validate ALS's robustness and state-of-the-art performance on homophilic graphs, heterophilic graphs, and long-range graph benchmarks, with strong baselines including recently studied Graph Transformers and Graph Mambas.

## 1. Introduction

The graph attention (GAT) mechanism (Velickovic et al., 2017; Brody et al., 2021) facilitates the soft selection of passing messages by assigning dynamic weights to edges based on the relevancies between adjacent nodes. It effectively captures data dependencies with more complex patterns than its non-attentive counterparts (Kipf & Welling, 2016) and has been widely adopted in various graph representation learning tasks (Huang & Carley, 2019; Kosaraju et al., 2019; Ma et al., 2020).

Recently, researchers have shown an increasing interest in graphs where data dependencies are both complex and distant (Dwivedi et al., 2022), such as chemical interaction modeling (Ying et al., 2021) and graph-based robotic controlling (Kurin et al., 2020). While stacking multiple GAT layers expands the receptive field (Chen et al., 2018) and enables a node to receive long-range messages from distant nodes, the number of stacked layers and the distance of message passing are still constrained by the emerging over-smoothing problem (Oono & Suzuki, 2019). Specifically, node representations tend to become too similar to one another, making them indistinguishable as messages are repeatedly passing. Moreover, the growing memory footprint and computational overhead often limit the number of GAT layers, particularly in training on large-scale graphs.

Previous studies (Wang et al., 2020a; Choi, 2022) have combined GAT with the Personalized PageRank (PPR) (Page et al., 1999) to resolve the over-smoothing problem. PPR is to propagate information while retaining a fraction of the initial representations (Chen et al., 2020), ensuring the existence of a converged and distinguishable representation for each node after repeatedly message passing (Gasteiger et al., 2018; Bojchevski et al., 2020; Chien et al., 2020; Roth & Liebig, 2022). However, applying GAT into every propagation step of PPR requires infinite memory footprint for gradient descent. Existing approaches have developed various truncated versions of PPR to bypass this challenge (Wang et al., 2020a; Choi, 2022), but resulted in compromised message passing distance.

To capture long-range and complex data dependencies in diverse graph-based applications without the drawbacks of increasing memory footprint and the over-smoothing problem, this work proposes Attentive Long-Short-range Message Passing (ALS). Our contributions are as follows:

- We derive an optimization algorithm for the infinite-step PPR and reduce the memory footprint from the traditional hop-wise backpropagation's infinity to a constant. The resulting Differentiable PPR (DPPR) allows ALS to extend the propagation distance to infinity

[1]Anonymous Institution, Anonymous City, Anonymous Region, Anonymous Country. Correspondence to: Anonymous Author <anon.email@domain.com>.

Preliminary work. Under review by the International Conference on Machine Learning (ICML). Do not distribute.

without necessitating an increase in memory footprint.

- We develop three acceleration techniques—conjugate gradient with symmetrized attentions, dominant eigenvector initialization, and adaptive batch-terminating—to expedite the computation of DPPR, achieving up to an 89.51% reduction in computation time compared to naive message passing. These techniques are also advantageous for enhancing the efficiency of existing PPR-based methods.

- We design a short-range message passing module to combine with DPPR, ensuring the robustness of ALS in learning under heterophily (Platonov et al., 2023). Node classification experiments on homophilic graphs, heterophilic graphs, and long-range graph benchmarks reveal that ALS outperforms state-of-the-art baselines, including the recently studied Transformers (Rampášek et al., 2022) and Mambas (Behrouz & Hashemi, 2024; Wang et al., 2024).

## 2. Background and Related Works

We assume that the adjacency matrix $\boldsymbol{A} \in \{0,1\}^{n \times n}$ represents a graph's connectivity, with element $A_{ij} = 1$ if graph node $i$ is connected to node $j$, and 0 otherwise. The feature matrix $\boldsymbol{X} \in \mathbb{R}^{n \times d}$ contains the node features, where the $i$-th row $\boldsymbol{x}_i$ is a feature vector associated with node $i$.

### 2.1. Graph Attentions

The attention mechanism, which dynamically weighs the influence of a part in an inputted sequence to another part, is the core innovation of Transformer (Vaswani et al., 2017). Supposing the initial representation of part $i$ is a $d$-dimensional row vector $\boldsymbol{x}_i$, the attention mechanism relies on three vectors derived from $\boldsymbol{x}_i$: $(\boldsymbol{q}_i, \boldsymbol{k}_i, \boldsymbol{v}_i) = \boldsymbol{x}_i \cdot (\boldsymbol{W}_q, \boldsymbol{W}_k, \boldsymbol{W}_v)$, where $\boldsymbol{W}_q, \boldsymbol{W}_k, \boldsymbol{W}_v \in \mathbb{R}^{d \times c}$ are optimizable parameters. The influencial score $s_{ij}$ of part $i$ to part $j$ is given by

$$s_{ij} = \frac{\boldsymbol{q}_i \cdot \boldsymbol{k}_j^T}{\sqrt{c}}, \tag{1}$$

where $\boldsymbol{q}_i$ and $\boldsymbol{k}_j$ are the query and key vectors derived from $\boldsymbol{x}_i$ and $\boldsymbol{x}_j$, respectively. Then, normalizing $\exp(s_{ij})$ within the context (the sequence) $N$ gets the attention weights

$$\tilde{A}_{ij} = \frac{\exp(s_{ij})}{D_i}, D_i = \sum_{j \in N} \exp(s_{ij}). \tag{2}$$

Finally, The representations $\boldsymbol{h}_i$ of part $i$ are obtained by gathering weighted information from all parts within the context, as $\boldsymbol{h}_i = \sum_{j \in N} \tilde{A}_{ij} \boldsymbol{v}_j$. The attention mechanism inspires many subsequent works in natural language processing (Kalyan

et al., 2021) due to its prominent performance and is soon applied in other domains such as computer vision (Han et al., 2020).

Graph attention networks (GAT) (Velickovic et al., 2017) has brought the attention mechanism into graph neural networks (Wu et al., 2020) to empower their adaptiveness and is later improved to GATv2 (Brody et al., 2021). The influencial score $s_{ij}$ of a graph node $i$ to its adjacent node $j$ is given by

$$s_{ij} = \sigma(\boldsymbol{q}_i + \boldsymbol{k}_j)\boldsymbol{v}^T, \tag{3}$$

where $\boldsymbol{v}$ is optimizable and $\sigma(\cdot)$ is a nonlinear function usually implemented as LeakyReLU (Maas et al., 2013) with its negative slope as 0.2. The key feature of GAT/GATv2 is that the attention context is decided by the graph structure: The attention context of node $i$ is its neighbourhood $N(i)$.

Researchers have also investigated another combination of graph and Transformer as Graph Transformers (GT) (Ying et al., 2021; Wu et al., 2021) where the attention context is the whole node set. By allowing to attend to all nodes, GT captures data dependencies between arbitrary node pairs, alleviating the over-smoothing problem in message passing. However, GT confronts with a new challenge of the over-globalizing problem (Xing et al., 2024) that useful information on near nodes are weakened. Besides, similar to the reliance of Transformer on positional encodings, which convey positional information of the processing part in the sequence, GT has a severe performance downgrade if it works without information about the graph structure (Dwivedi & Bresson, 2020). Therefore, many recent studies on GT (Shirzad et al., 2023a; Ma et al., 2023) located in a framework termed GraphGPS (Rampášek et al., 2022), which integrates structural encodings (SE, such as SAN (Kreuzer et al., 2021) and RWSE (Dwivedi et al., 2021)), message-passing neural networks (MPNN, such as GAT/GATv2), and GT together to obtain comprehensive representations.

Our proposed method (ALS) still aligns in the line of MPNN and is orthogonal to GT and SE, which means that SE can also augment the node features in ALS to boost its performance, and ALS can be an advanced candidate of the MPNN module in GraphGPS.

### 2.2. Personalized PageRank

Personalized PageRank (PPR) (Page et al., 1999) has been introduced in graph representation learning (Gasteiger et al., 2018) to resolve the over-smoothing problem (Oono & Suzuki, 2019) emerged in long-range message passing. Given initial representations $\boldsymbol{X}$, the PPR is defined as an iterative equation

$$\boldsymbol{Z}^{(k+1)} = (1-\alpha)\tilde{\boldsymbol{A}}\boldsymbol{Z}^{(k)} + \alpha\boldsymbol{X}, k = 0, 1, \ldots \tag{4}$$

where $\alpha \in (0, 1]$ is the teleport probability which helps every node to preserve its local information in the resulted representations $\boldsymbol{Z}^{(k+1)}$. The iteration converges exponentially to

$$\boldsymbol{Z}^{(\infty)} = \alpha \left( \boldsymbol{I} - (1-\alpha)\tilde{\boldsymbol{A}} \right)^{-1} \boldsymbol{X} \triangleq \Pi \boldsymbol{X}, \quad (5)$$

notated as the result of PPR$(\alpha, \tilde{\boldsymbol{A}}, \boldsymbol{X})$.

Various approaches have been proposed to leverage the long-range propagation capability of PPR. APPNP (Gasteiger et al., 2018), MAGNA (Wang et al., 2020a), and GPRGNN (Chien et al., 2020) iterate Equation 4 for $K$ steps starting from $\boldsymbol{Z}^{(0)} = \boldsymbol{0}$ and take $\boldsymbol{Z}^{(K)}$ as the final representations. However, the number $K$ of propagation steps is limited due to the increasing computation overhead. Additionally, when $\tilde{\boldsymbol{A}}$ is dynamically generated, the memory footprint for gradient computation grows linearly, further restricting $K$. PPRGo (Bojchevski et al., 2020) and PPRGAT (Choi, 2022) precompute $\Pi$ and utilize it as the weighted adjacency matrix in graph neural networks. To avoid the $O(n^2)$ complexity of $\Pi$, they approximate it using a sparsified version in which small elements are truncated. Given that an element is typically small when it measures the connection between two distant nodes in the original graph, long-range dependencies are likely to be ignored due to the sparsification. In summary, these approaches employ various truncated versions of PPR, which compromises the message passing distance.

Our work utilizes dynamic $\tilde{\boldsymbol{A}}$ and obtains the accurate $\boldsymbol{Z}^{(\infty)}$ by iterating Equation 4 for an unlimited number of steps till convergence. We derive an iterative algorithm from the implicit differentiation theory (Bai et al., 2019) to compute gradients without knowing intermediate representations, consuming only constant memory.

### 2.3. Implicit Differentiation

The implicit differentiation theory is adopted in implicit graph neural networks (IGNN) (Gu et al., 2020; Liu et al., 2021; Roth & Liebig, 2022; Liu et al., 2022; Chen et al., 2022b) to derive their backpropagation algorithms. IGNN outputs the equilibrium solution $\boldsymbol{H}^*$ of the following equation

$$\boldsymbol{H} = \sigma(\boldsymbol{Z}) = \sigma(\tilde{\boldsymbol{A}}\boldsymbol{H}\boldsymbol{W} + b_{\boldsymbol{\theta}}(\boldsymbol{X})), \quad (6)$$

where $\boldsymbol{W}$ and the network $b_{\boldsymbol{\theta}}$ are optimizable. This equation is solved using the naive iteration method like Equation 4 if the iteration is contractive. However, the equation may need a large number of iterations to meet a convergence, consuming a prohibitive memory footprint to store intermediate activations for gradient descent. Therefore, IGNN derives its backpropagation algorithm from the implicit differentiation theory. Specifically, gradients $\nabla_{\boldsymbol{H}^*}$ of $\boldsymbol{H}^*$ is given by

solving another equilibrium equation

$$\nabla_{\boldsymbol{H}} = \tilde{\boldsymbol{A}}^T \cdot (\nabla_{\boldsymbol{H}} \odot \frac{\partial \sigma(\boldsymbol{Z})}{\partial \boldsymbol{Z}}) \cdot \tilde{\boldsymbol{W}}^T + \frac{\partial \mathcal{L}(\boldsymbol{H})}{\partial \boldsymbol{H}}, \quad (7)$$

where $\mathcal{L}$ is the loss function. Then, $\nabla_{\boldsymbol{H}^*}$ is plugged into the chain rule to derive the gradients of $\boldsymbol{W}$ and $\boldsymbol{\theta}$. Due to the repeatedly injected $b_{\boldsymbol{\theta}}(\boldsymbol{X})$ during the forward iterations, IGNNs are also a series of methods for long-range message passing without the over-smoothing problem.

Our work ALS also derives its backpropagation algorithm from the implicit differentiation theory to achieve the constant memory footprint, but it has two differences from IGNNs. First, $\tilde{\boldsymbol{A}}$ in ALS is optimizable, showing more powerful adaptiveness in graph representation learning. Second, we develop three acceleration techniques for PPR, making ALS to have at least 3.67 times the speed of IGNN in experiments.

## 3. Methodology

To capture long-range and complex data dependencies within graphs, we propose **Attentive Long-Short-range message passing (ALS)**, as depicted in Figure 1.

ALS integrates attention mechanisms to generate the attentive transition matrix for message passing, thereby enhancing its capability in capturing complex data dependencies. The core innovations of ALS are the accelerated Differentiable Personalized PageRank (DPPR) operator and the short-range message passing module. The accelerated DPPR operator allows central nodes to gather information from distant nodes with optimized speed and a constant memory footprint. The short-range message passing module focuses on the diversified information within local neighbourhoods, aiding in learning under heterophily. Overall, the ALS layer can serve as a building block for constructing graph neural networks. It is characterized by its ability to capture long-range dependencies, and its robustness across both homophilic graphs and heterophilic graphs.

In the following subsections, we sequentially describe our algorithms for differentiating the Personalized PageRank (PPR) process, techniques to accelerate PPR, and the module of short-range message passing designed to handle heterophily.

### 3.1. DPPR: Differentiable Personalized PageRank

Due to the infinite memory footprint of PPR for error-backpropagation, previous studies have resorted to truncated versions of PPR, resulting in a compromise in their capabilities for long-range message passing. In contrast, we leverage the full capability of untruncated PPR by presenting a novel differentiation algorithm, which consumes only constant memory.

Figure 1. The architecture of the Attentive Long-Short-range message passing (ALS) layer (the middle part). The core innovations of ALS are **1)** the accelerated DPPR operator (the left lower part), which enables central nodes to gather information from distant nodes with optimized speed and a constant memory footprint, and **2)** the short-range message passing module (the left upper part), which helps ALS learning under heterophily. We construct a single-layered ALS network (the right upper part) and a multi-layered ALS network (the right lower part) for different experimental purposes.

The following theorem facilitates the differentiation of PPR, with its proof provided in Appendix A:

**Theorem 3.1.** *The gradients of*

$$Z = PPR(\alpha, \tilde{A}_\theta, V)$$

*can be obtained by another PPR process:*

$$\nabla_Z = PPR\left(\alpha, \tilde{A}_\theta^T, \left(\frac{1}{\alpha} \cdot \frac{\partial \mathcal{L}(Z)}{\partial Z}\right)\right),$$

*where $\tilde{A}_\theta$ is the attention matrix, $V$ is the inputted representations, and $\mathcal{L}(Z)$ is the loss.*

By plugging $\nabla_Z$ into the chain rule, we can compute the gradients of $V$ and $\tilde{A}_\theta$, thereby enabling gradient descent to optimize the PPR process.

$$\begin{cases} \nabla_V = \alpha \cdot \nabla_Z \\ \nabla_{\tilde{A}_\theta} = (1 - \alpha) \cdot A \odot (\nabla_Z \cdot Z^T) \end{cases}.$$

It is worth noting that we do not have to compute the dense matrix of $(\nabla_Z \cdot Z^T)$, which would have a complexity of $O(n^2)$. Instead, we compute the resulting non-zeros of $A \odot (\nabla_Z \cdot Z^T)$ edge by edge, with a complexity of $O(m)$, where $m$ is the number of edges.

The PPR optimized using this algorithm is termed Differentiable PPR (DPPR). We have encapsulated DPPR as a

PyTorch (Paszke et al., 2019) operator in our code [1]. Community researchers can replace their propagation operator with DPPR to expand the receptive field to infinity without increasing memory footprint.

### 3.2. Accelerated DPPR

In many cases, a relatively small $\alpha$ in PPR is necessary to obtain expressive representations (Boldi et al., 2007). However, small $\alpha$ can lead to a deterioration in convergence speed, especially when solving PPR with the naive iteration method (Gleich, 2015). Therefore, we develop three techniques to accelerate DPPR, which also benefit to the computational efficiency of existing PPR-based methods such as PPNP and MAGNA.

#### 3.2.1. CONJUGATE GRADIENT (CG) WITH SYMMETRIZED ATTENTIONS (SYMGAT)

Many previous works (Berkhin, 2005; Golub & Greif, 2006; Zhang et al., 2016) suggest treating the result $Z$ of $PPR(\alpha, \tilde{A}_\theta, V)$ as a solution to the linear system

$$\frac{1}{\alpha}\left(I - (1 - \alpha)\tilde{A}_\theta\right) Z = V \tag{8}$$

[1] https://anonymous.4open.science/r/ALS-A104

and solving it using the Krylov subspace method (Liesen & Strakos, 2013), such as GMRES and MINRES (Saad, 2003), which can converge rapidly even when $\alpha$ is small. However, the construction of the Krylov subspace may necessitate tens of times the memory footprint consumed by the entire node representations $\boldsymbol{V}$, making these implementations impractical for large-scale graphs.

To integrate the Krylov subspace method into practical graph representation learning, we replace $\tilde{\boldsymbol{A}}_{\boldsymbol{\theta}}$ in PPR with symmetrized attentions (Wang et al., 2021) defined as

$$\hat{A}_{ij} = \frac{\exp(s_{ij})}{\sqrt{D_i \odot D_j}},$$

where we utilize $\boldsymbol{W}_q = \boldsymbol{W}_k$ to ensure $s_{ij} = s_{ji}$. With the symmetrized attentions $\hat{\boldsymbol{A}}_{\boldsymbol{\theta}}$, a space-saving implementation of the Krylov subspace method, known as conjugate gradient (Hestenes et al., 1952), becomes feasible. This implementation consumes only twice the memory footprint of $\boldsymbol{V}$ to store the residuals and the conjugate vectors, making it suitable for large-scale graph processing.

### 3.2.2. DOMINANT EIGENVECTOR INITIALIZATION (EIGENINIT)

By default, both the naive iteration method and CG commence their iterations from $\boldsymbol{0}$, which leverages no information encoded in $\boldsymbol{A}_{\boldsymbol{\theta}}$.

On the contrary, we start our iterations from a better point $\bar{\boldsymbol{V}} = [\bar{\boldsymbol{v}}_1; \bar{\boldsymbol{v}}_2; \ldots; \bar{\boldsymbol{v}}_n]$, where $\bar{\boldsymbol{v}}_i$ is the projection of the $i$-row of $\boldsymbol{V}$ onto the dominant eigenvector of $\boldsymbol{A}_{\boldsymbol{\theta}}$. We notice that $\boldsymbol{1}$ is the dominant eigenvector of $\tilde{\boldsymbol{A}}_{\boldsymbol{\theta}}$ and $(\sqrt{D_1}, \sqrt{D_2}, \ldots, \sqrt{D_n})$ (defined in Equation 2) is the dominant eigenvector of the symmetrized $\hat{\boldsymbol{A}}_{\boldsymbol{\theta}}$. Both dominant eigenvalues are 1. Therefore, the residual after the first iteration of Equation 8 starting from $\bar{\boldsymbol{V}}$ is $\|\bar{\boldsymbol{V}} - \boldsymbol{V}\|$, which is not greater than the residual $\|\boldsymbol{V}\|$ of starting from $\boldsymbol{0}$, meaning that starting from $\bar{\boldsymbol{V}}$ is expected to converge in fewer iterations.

### 3.2.3. ADAPTIVE BATCH-TERMINATING (ADATERM)

We observe that Equation 8 is composed of $H \times C$ independent linear systems, which may converge at different speeds. Stopping iterations adaptively for linear systems that have already converged saves computation and accelerates our solvers.

Considering multi-heads attentions, the $n \times n$ attention matrix $\boldsymbol{A}_{\boldsymbol{\theta}}$ in our previous discussion becomes an $n \times n \times H$ tensor, where $H$ is the number of attention heads. Similarly, the input tensor $\boldsymbol{V}$ and the resulting tensor $\boldsymbol{Z}$ are both $n \times H \times C$, where $C$ is the number of representation channels in each attention head. There are $H \times C$ independent linear systems in Equation 8. As Algorithm 1 describes, after

---

**Algorithm 1** PPR with AdaTerm

**Input**: $f$ which iterates the naive iteration method or the conjugate gradient for one step, $\alpha$, the attention tensor $\boldsymbol{A}_{\boldsymbol{\theta}}$, the inputting tensor $\boldsymbol{V}$, the convergence tolerance $\epsilon$. **Output**: the PPR result $\boldsymbol{Z}$

1: $m_h := \boldsymbol{1} \in \{0, 1\}^H$ {indicators of active heads}
2: $m_c := \boldsymbol{1} \in \{0, 1\}^C$ {indicators of active channels}
3: $\boldsymbol{r} := \boldsymbol{0} \in \mathbb{R}^{H \times C}$ {linear system residuals}
4: $\boldsymbol{Z} := \boldsymbol{0}$
5: **repeat**
6:    $\boldsymbol{Z}' := \boldsymbol{Z}$
7:    $\boldsymbol{Z}[:, m_h, m_c] := f(\alpha, \boldsymbol{A}_{\boldsymbol{\theta}}[:, m_h], \boldsymbol{V}[:, m_h, m_c]$
8:    $\boldsymbol{r}[h, c] := \|\boldsymbol{Z}[:, h, c] - \boldsymbol{Z}'[:, h, c]\|, \forall h \le H, c \le C$
    {Terminate converged heads}
9:    **if** $\exists h \le H, \forall c \le C, \boldsymbol{r}[h, c] < \epsilon$ **then**
10:      $m_h[h] := 0$
11:    **end if**
    {Terminate converged channels}
12:    **if** $\exists c \le C, \forall h \le H, \boldsymbol{r}[h, c] < \epsilon$ **then**
13:      $m_c[c] := 0$
14:    **end if**
15: **until** $\boldsymbol{r}[h, c] < \epsilon, \forall h \le H, c \le C$
16: **return** $\boldsymbol{Z}$

---

every iteration step, we check if any linear system meets its stop condition for convergence. If the linear systems of all channels within an attention head have all converged, we stop iterating the entire head in subsequent iterations. Likewise, we stop iterating an entire channel if the linear systems of all heads along that channel have converged. This adaptive terminating improves the efficiency of our solvers by reducing unnecessary computation.

### 3.3. Short-Range Message Passing

In some graphs where adjacent nodes exhibit heterophilic characteristics, it is necessary to differentiate the processing of a node's own features from the features of its neighbours for improved performance (Zhu et al., 2020). To compensate for the DPPR operator's limitations in such diverse processing, we design an additional short-range message passing module for ALS. This module effectively handles heterophily without negatively impacting the performance on homophilic graphs.

An ALS layer with $K$ steps of short-range message passing outputs $\boldsymbol{H}$ as

$$\begin{cases} \boldsymbol{H} = \sigma(\boldsymbol{Z}^{(K)}) \\ \boldsymbol{Z}^{(k)} = (1 - \alpha)\boldsymbol{A}_{\boldsymbol{\theta}}\boldsymbol{Z}^{(k-1)} + \alpha\boldsymbol{X}\boldsymbol{W}_k, k = 1, 2, \ldots, K \\ \boldsymbol{Z}^{(0)} = \mathrm{DPPR}(\alpha, \boldsymbol{A}_{\boldsymbol{\theta}}, \boldsymbol{X}\boldsymbol{W}_0) \end{cases},$$

where $\sigma(\cdot)$ is a non-linear activator such as ReLU (Maas et al., 2013) or layer normalization.

$W_0, W_1, W_2, \ldots, W_K$ are optimizable parameters. The short-range message passing module is placed after DPPR. It propagates information in a manner akin to truncated PPR but applies different transformations $W_1, W_2, \ldots, W_K$ to information at different steps. When $K$ is a positive integer, short-range information about a node within its $K$-hops neighbourhood is processed separately by the diversified transformations to accommodate heterophily. On graphs without such heterophily, $W_1, W_2, \ldots, W_K$ can approximate $W_0$ to implicitly disable the short-range message passing module, thus avoiding any impact on the DPPR results.

## 4. Experiments

We conduct **node classification** experiments on a variety of graphs to elucidate the characteristics of our proposed methods and show the powerfulness of ALS. The datasets employed in our experiments are categorized as four groups. The first group is homophilic graphs, where nearby nodes share similar labels, including Amazon Computer, Amazon Photo (McAuley et al., 2015), Coauthor CS, Coauthor Physics (Wang et al., 2020b; Shchur et al., 2018), and WikiCS (Mernyei & Cangea, 2020). The second group is heterophilic graphs, where adjacent nodes tend to have different labels, including Roman Empire, Amazon Ratings, Minesweeper, Tolokers, and Questions (Platonov et al., 2023). The third group is the large-scale OGB-Arxiv and OGB-Products dataset (Hu et al., 2020). The last group is the long-range graph benchmarks, where labels strongly depend on long-range interactions between nodes, including PascalVOC-SP and COCO-SP (Dwivedi et al., 2022). [2] Details of the baselines and other experimental settings are in Appendix E.

### 4.1. Efficiency of DPPR on Memory Saving

In this section, we utilize the OGB-Arxiv dataset to conduct experiments aimed at validating the efficiency of DPPR in terms of memory saving.

Figure 2 visually depicts the memory footprints involved in optimizing a weighted version of the label propagation algorithm (Zhu et al., 2005), which propagates label information along edges. The edge weights in this implementation are dynamically computed based on node features (Wang & Leskovec, 2021). It is evident from the figure that as the number of propagation steps increases, the conventional hop-wise backpropagation method consumes an increasing amount of memory, whereas DPPR maintains a con-

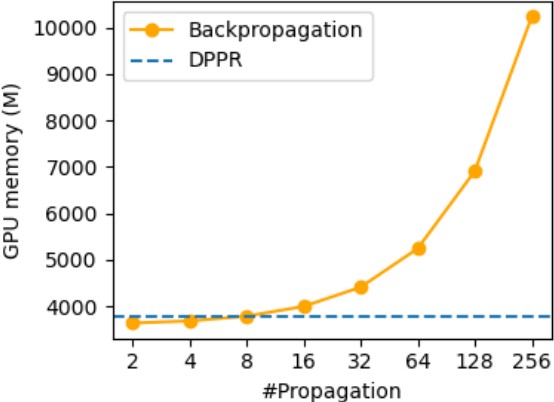

*Figure 2.* Memory footprint of different optimization algorithms on the OGB-Arxiv dataset

stant memory footprint. This is because DPPR obtains the gradients of its results by another PPR process, thereby consuming constant memory regardless of the number of propagations.

In conclusion, the DPPR operator demonstrates efficiency in memory saving compared to conventional GNNs.

### 4.2. Effectiveness of the Acceleration Techniques

In this section, we evaluate the effectiveness of our proposed acceleration techniques: CG with SymGAT, EigenInit, and AdaTerm, which are applied on a single-layered ALS.

We select IGNN (Gu et al., 2020) as a baseline for comparison due to its iterative nature, which is similar to ALS. The maximum number of iterations for IGNN and ALS is set to 300. Each model is trained for 20 epochs using an NVIDIA GeForce RTX 3060 GPU, and the time taken for each epoch is measured. The averaged training time per epoch is reported in Figure 3, with the experimented graphs sorted by their average node degrees and annotated as either homophilic or heterophilic. AsymGAT refers to the ALS using the original version of graph attentions (Equation 2).

As the figure illustrates, the speed of AsymGAT deteriorates significantly when $\alpha$ decreases. 'AsymGAT + AdaTerm' consistently reduces the training time of AsymGAT by about 10% to 20%. 'AsymGAT + EigenInit' [3] demonstrates negligible effects on homophilic graphs but is notably effective on heterophilic graphs. 'SymGAT + CG' is the fastest technique. Its speed deteriorates more slowly than AsymGAT when $\alpha$ decreases, and its efficiency is further improved as the graphs become denser. With the incorporation of all

---

[2]PascalVOC-SP and COCO-SP are 2 of the 5 proposed datasets from Dwivedi et al. (2022) used to assess a model's capability in capturing long-range dependencies. We exclude the other 3 datasets because they are recently found to be inadequate for this purpose (Tonshoff et al., 2023).

[3]While EigenInit accelerates both the forward and backward iterations in SymGAT, it only accelerates the forward iterations in AsymGAT, because the dominant eigenvector of $A_\theta^T$ is unknown.

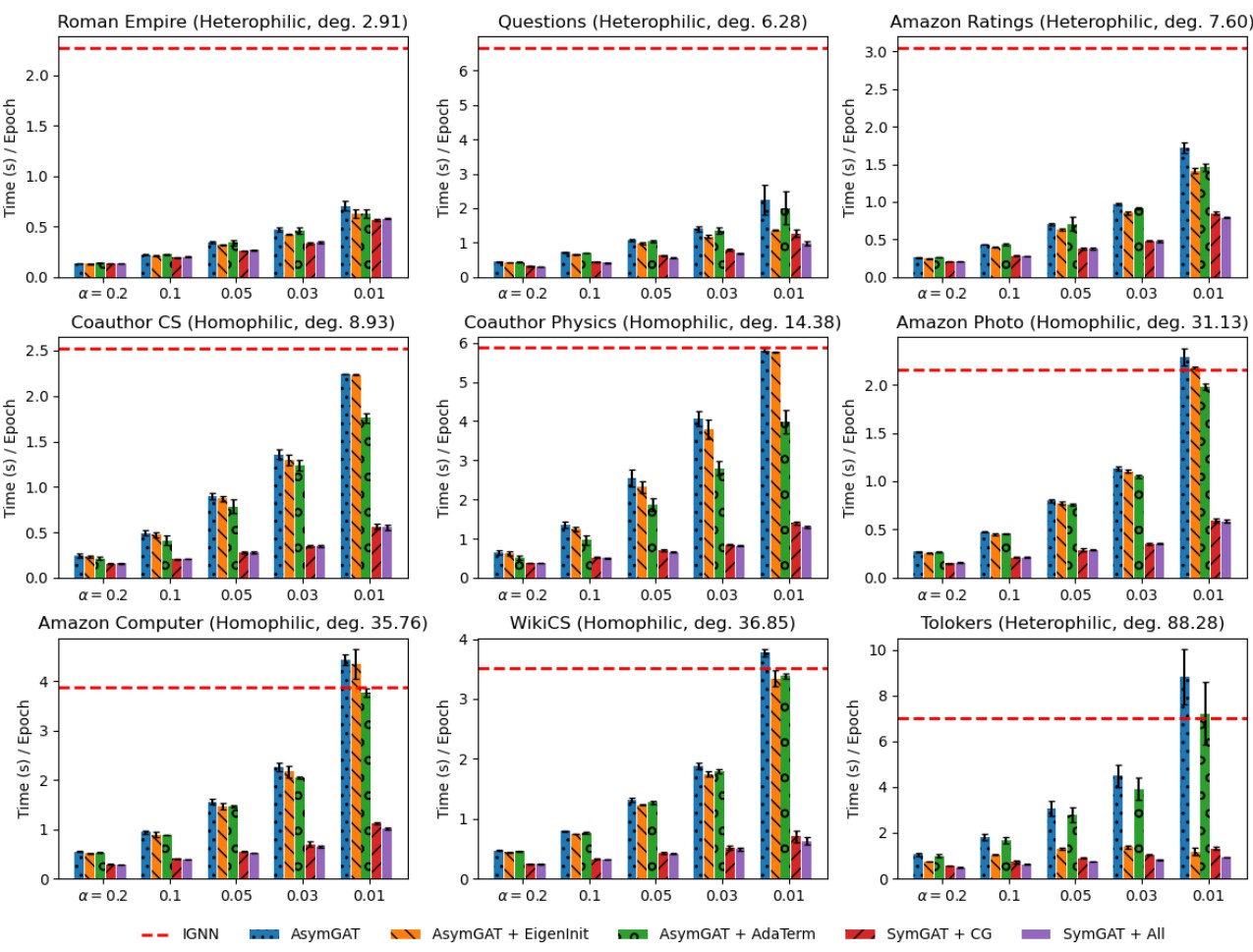

*Figure 3.* Averaged time for an epoch of training when different acceleration techniques involved

these acceleration techniques, 'SymGAT + All' reduces the training time by up to 89.51% (on Tolokers) compared to the naive iteration method (AsymGAT), and is at least 3.67 times as fast (on Amazon Photo) as IGNN.

In conclusion, all three techniques are effective in accelerating PPR. Their combination significantly reduces the computation time of ALS, making it notably faster than another iterative baseline (IGNN) even when $\alpha$ is very small.

### 4.3. Compare ALS With State-of-the-Art Baselines

In this section, we demonstrate the node classification performance of ALS on 14 graphs and compare ALS with state-of-the-art baselines, including message-passing neural networks (MPNN) and Graph Transformers (GT).

Table 1 presents the averaged accuracy scores of ALS and 16 competitive MPNNs, including GCN (Kipf & Welling, 2016), GraphSAGE (Hamilton et al., 2017), GAT (Velick-ovic et al., 2017), GAT-sep (Platonov et al., 2023), GC-NII (Chen et al., 2020), GPRGNN (Chien et al., 2020), APPNP (Gasteiger et al., 2018), PPRGo (Bojchevski et al., 2020), GGCN (Yan et al., 2021), OrderedGNN (Song et al., 2023), tGNN (Hua et al., 2022), H2GCN (Zhu et al., 2020), FSGNN (Maurya et al., 2022), GloGNN (Li et al., 2022), G2-GNN (Rusch et al., 2023), and LINKX (Lim et al., 2021). As indicated in the table, ALS achieves state-of-the-art performance scores on 11 out of 12 datasets. Even on the Questions dataset where ALS is not the best, it still achieves the second-best performance. These results underscore the efficacy of ALS in handling homophilic, heterophilic, and large graphs.

Table 2 presents the averaged accuracy scores of ALS and other baselines across 4 runs on long-range graph benchmarks. Methods in the table is grouped into two sections, the message-passing neural networks (MPNN) category and the 'SE + GT/Mamba + MPNN' category. Our ALS is

*Table 1.* Averaged accuracy scores and the standard deviations in 10 runs on homophilic graphs (the upper section), heterophilic graphs (the middle section), and large graphs (the lower section). In each section, the best score for each dataset is **bolded**. Detailed scores of baselines are in Appendix E.

| | Baselines | | **ALS** |
| | Rank-1 | Rank-2 | |
| --- | --- | --- | --- |
| Amazon Computer | 92.03±0.13 | 91.81±0.20 | **92.11±0.33** |
| Amazon Photo | 95.10±0.20 | 94.59±0.14 | **95.94±0.28** |
| Coauthor CS | 95.25±0.05 | 95.13±0.09 | **96.22±0.12** |
| Coauthor Physics | 97.07±0.05 | 97.00±0.08 | **97.54±0.07** |
| WikiCS | 79.01±0.68 | 78.87±0.11 | **80.97±0.75** |
| Roman Empire | 88.75±0.41 | 87.32±0.39 | **88.90±0.54** |
| Amazon Ratings | 53.63±0.39 | 52.74±0.83 | **54.10±0.37** |
| Minesweeper | 93.91±0.35 | 93.51±0.57 | **95.55±1.05** |
| Tolokers | 83.78±0.43 | 83.70±0.47 | **86.09±0.92** |
| Questions | **78.86±0.92** | 76.79±0.71 | 78.26±0.95 |
| OGB-Arxiv | 72.01±0.20 | 71.74±0.29 | **72.71±0.22** |
| OGB-Products | 79.76±0.59 | 79.45±0.59 | **81.40±0.24** |

*Table 2.* Averaged accuracy scores and the standard deviations of baselines, include MPNNs (the upper section) and their integrations with GT or Mamba (the lower section), in 4 runs on the long-range graph benchmarks. Each run has 60 hours of time budget and IGNN on COCO-SP runs Out of Time (OoT). In each section, the best score for each dataset is **bolded**, and the second best is underlined.

| | PascalVOC-SP | COCO-SP |
| --- | --- | --- |
| GCN | 20.78±0.31 | 13.38±0.07 |
| GINE | 27.18±0.54 | 21.25±0.09 |
| GatedGCN | 38.80±0.40 | 29.22±0.18 |
| IGNN | 21.66±0.39 | OoT |
| APPNP | 31.24±0.47 | 22.05±0.09 |
| GPRGNN | 37.47±0.81 | 27.46±0.47 |
| **ALS** (as MPNN) | **39.59±0.61** | **30.08±0.37** |
| NAGphormer | 40.06±0.61 | 34.58±0.70 |
| GraphGPS (original) | 37.48±1.09 | 34.12±0.44 |
| Exphormer | 39.60±0.27 | 34.30±0.08 |
| Graph Mamba | 43.93±1.12 | **39.74±1.01** |
| GraphGPS (reassessed) | 44.40±0.65 | 38.84±0.55 |
| **ALS** (in GraphGPS) | **44.93±0.75** | 39.23±0.65 |

based on pure message passing and falls under the MPNN category, with baselines GCN, GINE (Hu et al., 2019), GatedGCN (Bresson & Laurent, 2017), IGNN (Gu et al., 2020), APPNP (Gasteiger et al., 2018), and GPRGNN (Chien et al., 2020). [4] The second category refers to integrations of structural encodings (SE), graph transformers (GT) or Mamba (Gu & Dao, 2023), and MPNN. We integrate ALS into GraphGPS (Rampášek et al., 2022), a framework of 'SE + GT + MPNN', as its MPNN module to compare with other integrated baselines, including NAGphormer (Chen et al., 2022a), GraphGPS (the original one) (Rampášek et al., 2022), Exphormer (Shirzad et al., 2023b), Graph Mamba (Behrouz & Hashemi, 2024), and GraphGPS (the reassessed one) (Tonshoff et al., 2023). As shown in the table, ALS only slightly lags behind Graph Mamba on the COCO-SP dataset, while demonstrating its superiority in capturing long-range dependencies compared with other MPNNs and other 'SE + GT + MPNN' integrations.

In conclusion, ALS consistently exhibits state-of-the-art performance on homophilic graphs, heterophilic graphs,

---

[4]Although there exists other newer implicit and PageRank-based GNNs, such as EIGNN (Liu et al., 2021), LazyGNN (Xue et al., 2023), PPRGo (Bojchevski et al., 2020), and PPRGAT (Choi, 2022), these approaches are not included in the baselines because they necessitate preprocessing of the underlying graph prior to training. For instance, EIGNN decomposes the adjacency matrix, LazyGNN partitions the graph, and both PPRGo and PPRGAT precompute the PageRank matrix. Consequently, training them with dynamic batches of subgraphs, as is the case in PascalVOC-SP and COCO-SP, consumes expensive time due to their repeated preprocessing.

and long-range graph benchmarks, and demonstrates competitive performance on large graphs, thus highlighting its effectiveness in handling diverse graph structures.

### 4.4. Other Experiments

Except for the previous experiments, we have also validated the efficiency of DPPR in terms of parameter utilization in Appendix B and the robustness of the short-range message passing module on handling different graphs in Appendix C. Comprehensive ablation studies on ALS, such as how important the attention is, are also conducted in Appendix D.

## 5. Conclusion

In this work, we introduce Attentive Long-Short-range message passing (ALS), a graph representation learning method that is characterized by its long-range receptiveness and robustness under heterophily. The pivotal innovation of ALS lies in its differentiable and efficient PPR operator, which enables the practical integration of PPR and GAT due to its constant memory footprint and accelerated computations. Extensive experiments have corroborated the superiority of ALS on homophilic graphs, heterophilic graphs, large graphs, and long-range graph benchmarks, underscoring the efficacy of our method in diverse graph-based applications.

## Impact Statement

This paper presents work whose goal is to advance the field of Machine Learning. There are many potential societal consequences of our work, none which we feel must be specifically highlighted here.

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

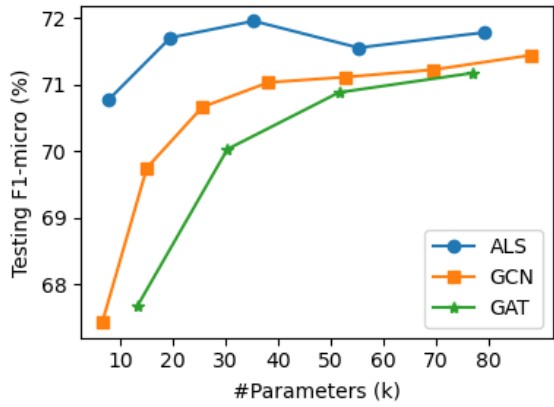

*Figure 4.* Parameters and accracy scores for different models

## A. Proof of Theorem 3.1

*Proof.* We rewrite PPR (Equation 5)

$$Z = \alpha \left( I - (1 - \alpha)\tilde{A}_\theta \right)^{-1} V$$

into the form of IGNN (Equation 6)

$$Z = \tilde{A}_\theta Z \cdot ((1 - \alpha)I) + (\alpha \cdot V)$$

and then derive the backpropagation algorithm as a corollary of Equation 7:

$$\nabla_Z = \tilde{A}_\theta^T \cdot \nabla_Z \cdot (1 - \alpha) + \frac{\partial \mathcal{L}(Z)}{\partial Z}$$
$$= (1 - \alpha)\tilde{A}_\theta^T \cdot \nabla_Z + \alpha \cdot \left( \frac{1}{\alpha} \cdot \frac{\partial \mathcal{L}(Z)}{\partial Z} \right)$$
$$= \text{PPR}(\alpha, \tilde{A}_\theta^T, \left( \frac{1}{\alpha} \cdot \frac{\partial \mathcal{L}(Z)}{\partial Z} \right)).$$

$\square$

## B. Efficiency of DPPR on Parameter Utilization

In this section, we utilize the OGB-Arxiv dataset to conduct experiments aimed at validating the efficiency of DPPR in terms of parameter utilization.

Figure 4 presents the performance scores of a 3-layered GCN, a 3-layered GAT, and a single-layered ALS with varying numbers of parameters. As depicted, ALS can achieve comparable performance to GCN and GATv2 with only about 20% parameters. This is because ALS decouples its propagations (Wu et al., 2019) and uses DPPR to expand the receptive field, thereby introducing no additional parameters rather than stacking graph convolutional layers.

In conclusion, the DPPR operator demonstrates efficiency in parameter utilization compared to conventional GNNs.

## C. Short-Range Message Passing for Heterophily

In this section, we validate the robustness of the short-range message passing module on homophilic graphs and heterophilic graphs.

Figure 5 visualizes the accuracy scores of a single-layered ALS with varying hops $K$ of short-range message passing. As illustrated in the figure, the short-range message passing module has negligible impact on the node classification performance

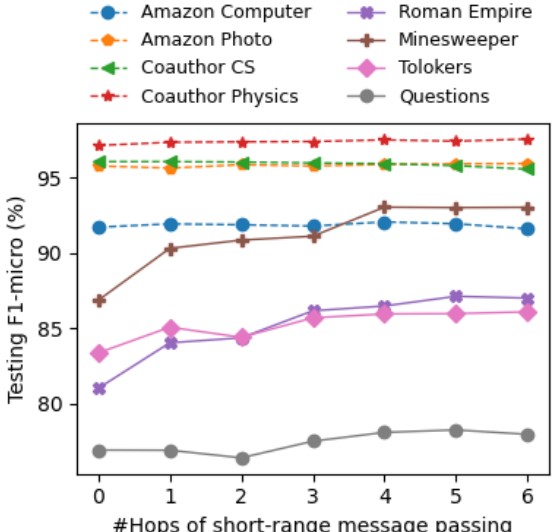

*Figure 5.* Averaged accuracy scores in 10 runs with different hops of short-range message passing on homophilic graphs (dashed lines) and heterophilic graphs (solid lines)

on homophilic graphs. However, in heterophilic graphs, the inclusion of this module significantly improves accuracy, and the improvement becomes more pronounced as $K$ increases.

In conclusion, the short-range message passing module aids ALS in better handling heterophily and has no adverse effect on homophilic graphs.

## D. Ablation Studies on ALS

In this section, we conduct ablation studies on ALS to gain a comprehensive understanding and address the following three questions:

- **Q1**: How powerful is DPPR when combined with attention weights?

- **Q2**: Does the symmetrization of GAT degrade its performance?

- **Q3**: Since the propagation range of ALS is infinite, is it necessary to stack multiple ALS layers?

By constraining hyperparameters, we derive the following variations of ALS:

- **NoGAT** represents ALS with a constant $\tilde{A}$, which is either the row-normalized or symmetrically-normalized adjacency matrix.

- **AsymGAT** employs the original version of GAT (Equation 2), with its influence scores computed by either Equation 1 or Equation 3.

- **SymGAT** is the symmetrized version of AsymGAT.

- **MultiLayered** refers to the multi-layerd ALS illustrated in Figure 1.

We present the accuracy scores for 10 runs of each ALS variation across 10 datasets in Figure 6. For **Q1**, the incorporation of attention weights significantly enhances the classification performance. NoGAT performs weaker than its attentive counterparts (AsymGAT and SymGAT) on 8 out of 10 graphs and only slightly better on Amazon Ratings and Tolokers. For **Q2**, SymGAT lags behind AsymGAT on 3 out of 5 homophilic graphs but surpasses AsymGAT on 3 out of 5 heterophilic

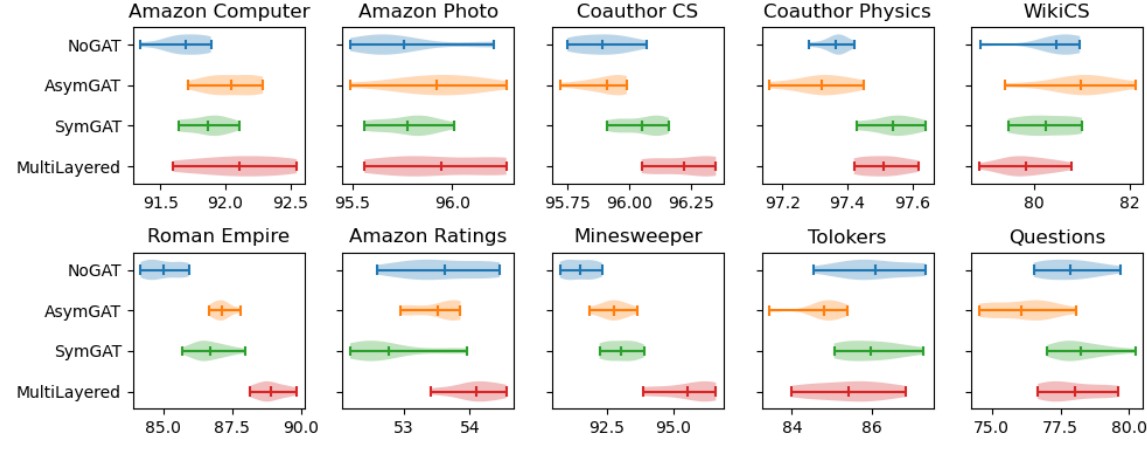

*Figure 6.* Testing F1-micro scores (%, the horizontal axes) in 10 runs of four ALS variations. The vertical line in the centre of each violin plot represents the average score.

*Table 3.* Statistics of datasets used in our experiments

|  | #Graphs | Homophily (%) | #Nodes | #Edges | Mean Deg. | Features | Classes |
|---|---|---|---|---|---|---|---|
| Amazon Computer | 1 | 68.23 | 13752 | 245861 | 35.76 | 767 | 10 |
| Amazon Photo | 1 | 78.50 | 7650 | 119081 | 31.13 | 745 | 8 |
| Coauthor CS | 1 | 78.45 | 18333 | 81894 | 8.93 | 6805 | 15 |
| Coauthor Physics | 1 | 87.24 | 34493 | 247962 | 14.38 | 8415 | 5 |
| WikiCS | 1 | 57.90 | 11701 | 216123 | 36.85 | 300 | 10 |
| Roman Empire | 1 | -4.68 | 22662 | 32927 | 2.91 | 300 | 18 |
| Amazon Ratings | 1 | 14.02 | 24492 | 93050 | 7.60 | 300 | 5 |
| Minesweeper | 1 | 0.94 | 10000 | 39402 | 7.88 | 7 | 2 |
| Tolokers | 1 | 9.26 | 11758 | 519000 | 88.28 | 10 | 2 |
| Questions | 1 | 2.07 | 48921 | 153540 | 6.28 | 301 | 2 |
| OGB-Arxiv | 1 | 58.92 | 169343 | 1166243 | 13.77 | 128 | 40 |
| OGB-Products | 1 | 80.76 | 2449029 | 61859140 | 50.52 | 100 | 47 |
| PascalVOC-SP | 11355 | — | 5443545 | 30777444 | 5.65 | 14 | 21 |
| COCO-SP | 123286 | — | 58793216 | 332091902 | 5.65 | 14 | 81 |

graphs. This suggests that symmetrizing GAT has no noticeable negative effect on performance. For **Q3**, there is no observable difference between single-layered and multi-layered ALS on 6 out of 10 graphs, except Coauthor CS, Roman Empire, Amazon Ratings, and Minesweeper. This indicates that the powerful propagation capability of ALS is sufficient for most cases. Stacking multiple ALS layers can serve as a fallback strategy to enhance performance when needed.

## E. Experimental Details

### E.1. Datasets

Table 3 details the 13 datasets employed in our experiments. The first five datasets are characterized as homophilic graphs (Shchur et al., 2018; Mernyei & Cangea, 2020). The subsequent five datasets are categorized as heterophilic graphs (Platonov et al., 2023). We incorporate the adjusted homophily (Platonov et al., 2023) to quantify how homophilic a graph is. As we can see, heterophilic graphs are all with lower homophily scores. In our experiments, the large-scale OGB-Arxiv and OGB-Products datasets (Hu et al., 2020) serves as benchmarks to evaluate the efficiency and scalability of the proposed methods. PascalVOC-SP and COCO-SP are recognized as long-range graph benchmarks (Dwivedi et al., 2022), where each graph in these benchmarks possesses an average shortest path length exceeding 10 and an average diameter

*Table 4.* Averaged accuracy scores and the standard deviations in 10 runs on homophilic (the upper section) and heterophilic graphs (the lower section). In each section, the best score for each dataset is **bolded**, and the second best is underlined.

| | Amazon Computer | Amazon Photo | Coauthor CS | Coauthor Physics | WikiCS |
|---|---|---|---|---|---|
| GCN | 89.65±0.52 | 92.70±0.20 | 92.92±0.12 | 96.18±0.07 | 77.47±0.85 |
| GraphSAGE | 91.20±0.29 | 94.59±0.14 | 93.91±0.13 | 96.49±0.06 | 74.77±0.95 |
| GAT | 90.78±0.13 | 93.87±0.11 | 93.61±0.14 | 96.17±0.08 | 76.91±0.82 |
| GCNII | 91.04±0.41 | 94.30±0.20 | 92.22±0.14 | 95.97±0.11 | 78.68±0.55 |
| GPRGNN | 89.32±0.29 | 94.49±0.14 | 95.13±0.09 | 96.85±0.08 | 78.12±0.23 |
| APPNP | 90.18±0.17 | 94.32±0.14 | 94.49±0.07 | 96.54±0.07 | 78.87±0.11 |
| PPRGo | 88.69±0.21 | 93.61±0.12 | 92.52±0.15 | 95.51±0.08 | 77.89±0.42 |
| GGCN | 91.81±0.20 | 94.50±0.11 | 95.25±0.05 | 97.07±0.05 | 78.44±0.53 |
| OrderedGNN | 92.03±0.13 | 95.10±0.20 | 95.00±0.10 | 97.00±0.08 | 79.01±0.68 |
| tGNN | 83.40±1.33 | 89.92±0.72 | 92.85±0.48 | 96.24±0.24 | 71.49±1.05 |
| **ALS** | **92.11±0.33** | **95.94±0.28** | **96.22±0.12** | **97.54±0.07** | **80.97±0.75** |

| | Roman Empire | Amazon Ratings | Minesweeper | Tolokers | Questions |
|---|---|---|---|---|---|
| GCN | 73.69±0.74 | 48.70±0.63 | 89.75±0.52 | 83.64±0.67 | 76.09±1.27 |
| GraphSAGE | 85.74±0.67 | 53.63±0.39 | 93.51±0.57 | 82.43±0.44 | 76.44±0.62 |
| GAT-sep | 88.75±0.41 | 52.70±0.62 | 93.91±0.35 | 83.78±0.43 | 76.79±0.71 |
| H2GCN | 60.11±0.52 | 36.47±0.23 | 89.71±0.31 | 73.35±1.01 | 63.59±1.46 |
| GPRGNN | 64.85±0.27 | 44.88±0.34 | 86.24±0.61 | 72.94±0.97 | 55.48±0.91 |
| FSGNN | 79.92±0.56 | 52.74±0.83 | 90.08±0.70 | 82.76±0.61 | **78.86±0.92** |
| GloGNN | 59.63±0.69 | 36.89±0.14 | 51.08±1.23 | 73.39±1.17 | 65.74±1.19 |
| GGCN | 74.46±0.54 | 43.00±0.32 | 87.54±1.22 | 77.31±1.14 | 71.10±1.57 |
| OrderedGNN | 77.68±0.39 | 47.29±0.65 | 80.58±1.08 | 75.60±1.36 | 75.09±1.00 |
| G2-GNN | 82.16±0.78 | 47.93±0.58 | 91.83±0.56 | 82.51±0.80 | 74.82±0.92 |
| tGNN | 79.95±0.75 | 48.21±0.53 | 91.93±0.77 | 70.84±1.75 | 76.38±1.79 |
| **ALS** | **88.90±0.54** | **54.10±0.37** | **95.55±1.05** | **86.09±0.92** | 78.26±0.95 |

exceeding 27.

For the homophilic graphs, we partition the nodes into training (60%), validation (20%), and testing (20%) sets, aligning with the approach outlined in Shirzad et al. (2023a). For the other graphs, we utilize the default data splits provided alongside the original datasets.

### E.2. Baselines

We have conducted extensive evaluations of ALS in Table 4 and Table 5, which are summarized as Table 1 in Section 4.3. In the tables, we compare ALS with more than 10 competitive graph neural networks, including GCN (Kipf & Welling, 2016), GraphSAGE (Hamilton et al., 2017), GAT (Velickovic et al., 2017), GAT-sep (Platonov et al., 2023), GCNII (Chen et al., 2020), GPRGNN (Chien et al., 2020), APPNP (Gasteiger et al., 2018), PPRGo (Bojchevski et al., 2020), GGCN (Yan et al., 2021), OrderedGNN (Song et al., 2023), tGNN (Hua et al., 2022), H2GCN (Zhu et al., 2020), FSGNN (Maurya et al., 2022), GloGNN (Li et al., 2022), G2-GNN (Rusch et al., 2023), and LINKX (Lim et al., 2021). To ensures that the accuracy scores of the baselines being compared are produced by well-tuned models, we retrieve scores of baselines for heterophilic and large graphs from Platonov et al. (2023), and for homophilic graphs from Deng et al. (2024).

In the upper section of Table 2, we compare ALS with MPNNs including GCN, GINE (Hu et al., 2019), GatedGCN (Bresson & Laurent, 2017), IGNN (Gu et al., 2020), APPNP (Gasteiger et al., 2018), and GPRGNN (Chien et al., 2020). In the lower section of Table 2, we integrate ALS into a reassessed version of GraphGPS (Tonshoff et al., 2023) and compare it with other 'SE + GT/Mamba + MPNN' methods, including NAGphormer (Chen et al., 2022a), GraphGPS (Rampášek et al., 2022), Exphormer (Shirzad et al., 2023b), and Graph Mamba (Behrouz & Hashemi, 2024; Wang et al., 2024). NAGphormer utilizes a Hop2Token module to encode information from local neighbourhoods based on message passing before applying a

*Table 5.* Averaged accuracy scores and the standard deviations in 10 runs on large graphs. The best score for each dataset is **bolded**, and the second best is underlined.

|        | OGB-Arxiv | OGB-Products |
|--------|-----------|--------------|
| GCN    | 71.74±0.29 | 75.64±0.21 |
| GAT    | 72.01±0.20 | 79.45±0.59 |
| GPRGNN | 71.10±0.12 | 79.76±0.59 |
| LINKX  | 66.18±0.33 | 71.59±0.71 |
| **ALS** | **72.71±0.22** | **81.40±0.24** |

GT backbone. GraphGPS is a framework that integrates structural encodings, MPNN, and GT to obtain comprehensive node representations. Exphormer enhances the GT module in GraphGPS with two novel sparse attention mechanisms: virtual global nodes and expander graphs. Graph Mamba replaces the GT module in GraphGPS with its long-range dependencies capturing modules based on random walks. The scores of baselines are sourced from previous works (Rampášek et al., 2022; Tonshoff et al., 2023; Behrouz & Hashemi, 2024; Wang et al., 2024), including their original papers and leaderboards of the respective datasets.

### E.3. Settings

As illustrated in Figure 1, we have implemented two networks using the ALS layer as a building block. The single-layered ALS is composed of an optional multi-layered perceptron (MLP) with 1 to 2 layers, an ALS layer, and a linear predictor for node classification. The multi-layered ALS is implemented like the multi-layered GCN model described in Platonov et al. (2023), with the exception that we replace the GCNConv layers with our ALS layers. In detail, the multi-layered ALS consists of a linear encoder, $L$ residual blocks, and a linear predictor. Each residual block is equipped with a skip connection (He et al., 2015) and comprises a layer normalization, an ALS layer, and a two-layered MLP.

Unless otherwise specified, we employ the Adam optimizer (Kingma & Ba, 2014) to train models with a learning rate of $r = 0.01$. The training process is limited to a maximum of 1000 epochs and employs an early stopping strategy to halt training if the performance on the validation set stagnates for 100 consecutive epochs. The default Dropout probability is set to $p = 0$.

In the following subsections, we describe the experimental settings for all figures and tables.

### E.3.1. FIGURE 2

To study the memory footprint of differentiable PPR (DPPR), We implement the weighted label propagation algorithm to propagate label information with a dynamic transition matrix as

$$\tilde{\boldsymbol{A}} = \sigma((\boldsymbol{x}_i \boldsymbol{W} + \boldsymbol{b}) \cdot \boldsymbol{B} \cdot (\boldsymbol{x}_j \boldsymbol{W} + \boldsymbol{b})^T),$$

where $\sigma$ is the row-wise Softmax function, $\boldsymbol{W}, \boldsymbol{b}$ and $\boldsymbol{B}$ are optimizable. The $\alpha$ for label propagation is set to 0.2.

### E.3.2. FIGURE 3

To assess the speed of the accelerated DPPR, we train IGNN and ALS with various acceleration techniques for 20 epochs and report the averaged training time per epoch. The maximum number of iterations is set to 300. ALS is configured with 16 attention heads, each of which is 16-dimensional. Short-range message passing is disabled for this experiment.

### E.3.3. TABLE 1, TABLE 5, FIGURE 5, AND FIGURE 6

We fine-tune ALS on homophilic, heterophilic, and large graphs using Optuna (Akiba et al., 2019) to determine the optimal number of ASL layers $L \in \{1, 2, 3, 4, 5\}$, the number of attention heads $H \in \{1, 2, \ldots, 10\}$, $\alpha \in [0.05, 0.8]$, the number of hops for short-range message passing $K \in \{0, 1, 2, \ldots, 6\}$, whether to apply attention weights, whether to symmetrize attentions, the way to compute influential scores (Equation 1 or Equation 3), the Dropout probability $p \in \{0, 0.1, 0.2, \ldots, 0.7\}$, and the learning rate $r \in \{0.01, 0.005, 0.001\}$. The results are shown in Table 1. During the fine-tuning, an adequate number of trials for ablation studies are conducted to generate Figure 6. Trials with $L = 1$ is

grouped by $K$ to study the effect of short-range message passing in Figure 5.

### E.3.4. TABLE 2

We inherit the code of GatedGCN and GraphGPS from Tonshoff et al. (2023) with few modifications to conduct our experiments on the long-range graph benchmarks. Specifically, we replace the GatedGCN layer with our ALS layer, maintaining the number of layers $L$ and the hidden dimensions $d$ unchanged. The number of attention heads $H$ is adjusted bellow 6 to ensure that the hidden dimensions $d$ is evenly divided by the heads $H$. The hops of short-range message passing $K$ is fixed to 2 to comply with the 500k parameter budget. For ALS as MPNN, no structural encodings are enabled. For ALS in GraphGPS, we use LapPE (Dwivedi & Bresson, 2020) to augment node features. To avoid tuning $\alpha$, we implement it as a $(1, H, C)$-sized tensor activated by the Sigmoid function and optimize it with its gradients computed as:

$$\nabla_\alpha = -\tilde{A}_\theta Z \odot \nabla_Z = -\frac{Z - \alpha V}{1 - \alpha} \odot \nabla_Z.$$

Thus, we only fine-tune the formula for influential scores (Equation 1 or Equation 3) and whether to symmetrize attentions. Other configurations, such as the optimizer and the maximum number of epochs, remain unchanged.

For baselines, including IGNN, APPNP, and GPRGNN, we also reuse the configuration file for GatedGCN but replace the GatedGCN layer with the evaluated baseline. We maintain the number of layers $L = 10$ for PascalVOC-SP and $L = 6$ for COCO-SP, respectively. The hidden dimensions $d$ is adjusted as large as possible within the 500k parameter budget. The hyperparameter $\alpha$ for APPNP and GPRGNN is tuned in the scope of $\{0.15, 0.5, 0.85\}$.

### E.3.5. FIGURE 4

We illustrate the parameters and accuracy scores for 3-layered GCN, 3-layered GAT, and the single-layered ALS. ALS is with $\alpha = 0.15$ and without short-range message passing. The number of attention heads is fixed to be 8 in both GAT and ALS. The probability of Dropout is $0.5$. Each model is run for 4 times and the average accuracy score is reported.

