# OpenReview forum: "ALS: Attentive Long-Short-Range Message Passing"
_ICML.cc/2025/Conference — Submitted to ICML 2025_

### Official Review · Reviewer_69P3 · 2025-02-16

**Overall Recommendation:** 2

**Summary:**

The paper presents Attentive Long-Short-range Message Passing to handle long-range dependencies while avoiding excessive memory usage and the over-smoothing problem

**Claims And Evidence:**

The authors conduct extensive experiments on 14 datasets, covering homophilic, heterophilic, and long-range graph benchmarks. In addition, they compare their method with recent algorithms such as GAT, APPNP, and IGNN. Lastly, the ablation studies show the impact of key components, including DPPR, attention mechanisms, and acceleration techniques

**Essential References Not Discussed:**

N/A

**Experimental Designs Or Analyses:**

The experiments cover diverse scenarios including homophily and heterophily. In addition, memory efficiency and training time are empirically validated

**Methods And Evaluation Criteria:**

The paper uses following metrics for evaluation: accuracy and F1 scores

**Other Comments Or Suggestions:**

N/A

**Other Strengths And Weaknesses:**

Strengths
* The acceleration technique reduces computation time by up to 89.51%
* ALS is compared against well-established GNN models such as GAT, APPNP, and IGNN
* Soundness of experimental designs, where the chosen datasets (e.g., Amazon, OGB, COCO-SP) are widely used in GNN research

Weaknesses
* The novelty of this paper lies in the Differentiable Personalized PageRank (DPPR) and accelerated training. However, computing only the non-zero gradients of PPR (Theorem 3.1) was already proposed in [1] (page 5, Section B). Additionally, the contribution of long-short-range message passing is incremental, as it was introduced in [2]. While long-short-range message passing may be effective for both homophilic and heterophilic graphs, the proposed algorithm is merely a simple integration of several existing methods, which significantly limits its novelty
  * [1]: Efficient Algorithms for Personalized PageRank Computation: A Survey
  * [2]: Long-short-range message-passing: A physics-informed framework to capture non-local interaction for scalable molecular dynamics simulation
* It would be better to demonstrate the effectiveness of the proposed method on widely used homophilic datasets (Cora, Citeseer, PubMed) and heterophilic datasets (Actor, Chameleon, Squirrel)

**Questions For Authors:**

Could you elaborate on the above weaknesses?

**Relation To Broader Scientific Literature:**

The proposed PPR and disjoint message-passing are widely used in literature

**Theoretical Claims:**

There seem to be no issues with the theoretical claims

---

> ### Author Rebuttal · Authors · 2025-03-27
>
> ## Response to concerns regarding novelty
>
> We have carefully examined the references you kindly pointed out and would like to clarify several aspects:
>
> > computing only the non-zero gradients of PPR (Theorem 3.1) was already proposed in [1]
>
> We respectfully note that the computation of non-zeros in $A \odot (\nabla_Z \cdot Z^T)$ at line 215 is not included in Theorem 3.1.
> So **even if** similar gradient computation methods exist, this does not diminish the novelty of Theorem 3.1, which demonstrates how to optimize a PPR process iterated to convergence - a fundamentally different approach from existing PPR-based methods with truncated iterations (e.g., PPNP and MAGNA).
>
> Moreover, the non-zero computation you mentioned in [1] actually refers to computing $P \hat \pi_s^{(L)}$.
> It is simply a forward-pass propagation, analogous to computing $A X$ in our framework.
> This differs significantly from our edge-wise computation of $A \odot (\nabla_Z \cdot Z^T)$.
>
> > "the contribution of long-short-range message passing is incremental"
>
> We are afraid that the reference [2] shares only method name similarity with our approach.
> Their method addresses molecular dynamics simulation in a two-level graph structure (atom-level many-body interations and molecule-level long-range interactions), whereas our work focuses on single-graph learning with distance-aware message passing.
> And heterophily is not considered in [2].
>
> > "the proposed algorithm is merely a simple integration of several existing methods"
>
> Due to the previous explanations, **we respectfully disagree with your assessment.**
> However, we greatly appreciate your positive comments on other aspects of our work.
>
> * [1] Efficient Algorithms for Personalized PageRank Computation: A Survey
> * [2] Long-short-range message-passing: A physics-informed framework to capture non-local interaction for scalable molecular dynamics simulation
>
> ## Response to concerns on datasets
>
> We sincerely appreciate your suggestion to evaluate ALS on additional datasets.
>
> However, we would like to note that reference [3] identifies significant limitations with Cora, Citeseer and Pubmed datasets, particularly regarding fragile and misleading results from their data splits.
> These datasets also represent a narrow range of network types (all citation networks).
> We instead employ the Amazon Computer, Amazon Photo, Coauthor CS, and Coauthor Physics datasets recommended by [3] as more robust homophilic benchmarks.
>
> For heterophilic graphs, reference [4] demonstrates issues with traditional datasets (e.g., train-test leakage in Chameleon and Squirrel) and proposes new benchmarks - the five heterophilic datasets we adopted.
>
> While we would be happy to conduct additional evaluations on your suggested datasets, we believe our current selection better represents modern, rigorous benchmarking practices in graph learning research.
>
> * [3] Shchur O. Pitfalls of Graph Neural Network Evaluation. ArXiv 2018 (#Citation: 1620)
> * [4] Platonov O. A critical look at the evaluation of GNNs under heterophily: Are we really making progress? ICLR 2023

---

### Official Review · Reviewer_HfV2 · 2025-03-14

**Overall Recommendation:** 2

**Summary:**

Overall, the core contribution of ALS includes a differentiable personalized PageRank and a short-range message-passing module for effectiveness and efficiency consideration of graph deep learning. The experiments are extensive, and the results are competitive. The writing and the organization of the paper can be largely improved, some critical parts are not quite clear. Details are as follows.

**Claims And Evidence:**

Most claims are supported. Two major concerns are listed below.

1. In the abstract and introduction, oversmoothing is regarded as one of the motivations of the paper. It is not easy to trace the relation between oversmoothing and the proposed method and how it can alleviate this problem theoretically and empirically.

2. The derivation or proof of theoretical time complexity mentioned in the abstract is not easy to locate in the main body of the paper.

**Essential References Not Discussed:**

To the best of the reviewer's knowledge, there is no big problem in this part.

**Experimental Designs Or Analyses:**

Overall, the performance of the proposed method seems competitive.

In table 1, "baselines" are vague.

**Methods And Evaluation Criteria:**

Evaluation criteria are fair and reasonable.

**Other Comments Or Suggestions:**

Please see all above

**Other Strengths And Weaknesses:**

Additional weaknesses:

1. In Section 3.1, the theorem comes as a sudden, very limited formation or expression of DPPR is introduced before, which makes it very hard to follow. It would be better to give the formal definition and expression of DPPR before the theoretical analysis.

2. The Equation (1) seems problematic, the dimension is inconsistent. $\mathbf{X}$ is $\mathbb{R}^{n \times d}$, but $\mathbf{Z}$ seems to be $\mathbb{R}^{n \times c}$, where $d$ is the number of input features, and $c$ is the number of labels, no matter based on line 124 in the paper or the original APPNP paper.

3. [Optional] Baselines in Table 4 are a subset of the mentioned in Section 4.3, and Section 4.3 lacks the baselines after 2023.

**Questions For Authors:**

Please see all above

**Relation To Broader Scientific Literature:**

Graph deep learning framework is interesting and important to many real-world impactful applications. The scope and vision of the paper is good.

**Theoretical Claims:**

As mentioned above, the detailed derivation of theoretical analysis seems missing.

---

> ### Author Rebuttal · Authors · 2025-03-27
>
> We sincerely appreciate the reviewer's time and valuable feedback.
> We hope our following clarifications help the reviewer better appreciate the significance of our contributions, and we would be happy to provide any additional information that might assist in their evaluation.
>
> ## Oversmoothing
>
> > It is not easy to trace the relation between oversmoothing and the proposed method
>
> We sincerely appreciate this thoughtful observation regarding oversmoothing analysis.
> As demonstrated in prior work [1,2], the integration of Personalized PageRank (PPR) fundamentally addresses oversmoothing in graph neural networks.
> Since this relationship has been well-established in the literature, we did not show the alleviation of this problem in our manuscript but focused our experimental validation on the novel aspects of our approach.
>
> * [1] Klicpera J. Predict then Propagate: Combining neural networks with personalized pagerank for classification on graphs. ICLR, 2018
> * [2] Choi J. Personalized pagerank graph attention networks. ICASSP, 2022
>
> ## Complexity
>
> > The derivation or proof of theoretical time complexity mentioned in the abstract is not easy to locate
>
> We appreciate this careful reading.
> We would like to clarify that the complexity analysis mentioned in the abstract specifically refers to memory complexity rather than time complexity.
> We will ensure this distinction is made clearer in any subsequent revisions.
>
> ## Baselines
>
> > In table 1, "baselines" are vague.
>
> Thank you for this helpful feedback.
> Table 1 presents aggregated results comparing our method against all baselines.
> The "Rank-1" column indicates the highest-performing baseline result for each dataset, while "Rank-2" shows the second-best baseline performance.
> Complete baseline results are available in Appendix E for reference.
>
> > Baselines in Table 4 are a subset of the mentioned in Section 4.3
>
> Thank you for this observation.
> We have verified that baselines are correctly described.
> The second paragraph of Section 4.3 describes baselines for Table 1, which is the combination of Table 4 and Table 5.
> In those baselines, only LINKX does not show up in Table 4, but it is in Table 5.
> The third paragraph introduces baselines for Table 2.
> So these baselines does not have to show up in Table 4.
>
> > Section 4.3 lacks the baselines after 2023
>
> We appreciate this suggestion for contemporary comparisons.
> Our evaluation includes several recent works of 2024, namely Graph Mambas [3,4].
> While other baselines predate 2023, many of them are from the recent literature [5].
>
> * [3] Behrouz A. Graph mamba: Towards learning on graphs with state space models. KDD 2024
> * [4] Wang C. Graph-mamba: Towards long-range graph sequence modeling with selective state spaces. arXiv 2024
> * [5] Chenhui D. Polynormer: Polynomial-Expressive Graph Transformer in Linear Time. ICLR 2024
>
> ## Notations
>
> > In Section 3.1, the theorem comes as a sudden...
>
> We apologize for any confusion caused by the theorem's presentation.
> The notation PPR($\alpha$, A, X) is explicitly defined in line 117 of the manuscript.
> We will consider adding additional transitional text to improve the flow in future revisions.
>
> > The Equation (1) seems problematic, the dimension is inconsistent...
>
> We appreciate this careful examination of our equations.
> The dimensions are indeed consistent: $x_i \in \mathbb{R}^{1 \times d}$ (line 89), so $q_i, k_j \in \mathbb{R}^{1 \times c}$ and $s_{ij}$ is a scalar.
> $c$ is just another dimension, not the number of labels.
> $\mathbf{Z}$ is node representations (same dimensionality as $\mathbf{X} \in \mathbb{R}^{n \times d}$) rather than classification outputs.

---

### Official Review · Reviewer_Wndg · 2025-03-14

**Overall Recommendation:** 2

**Summary:**

This study introduces Attentive Long-Short-range message passing (ALS), which combines personalized PageRank to address over-smoothing and utilizes GAT for capturing complex data dependencies, significantly reducing memory footprint and computation time. Extensive experiments show that ALS achieves competitive or superior results compared to other baselines.

**Claims And Evidence:**

It is easy to follow. However, while the authors claim that the proposed method mitigates oversmoothing, the experimental results do not provide a detailed analysis.  Moreover, the results in some cases do not show significant improvement over existing methods, and additional experiments or clearer explanations are needed.

**Essential References Not Discussed:**

No

**Experimental Designs Or Analyses:**

1. While the algorithm's complexity is analyzed, it would be beneficial to also evaluate its memory usage with other baselines.

2. Although the proposed method is designed to capture long-range information, it does not achieve significant improvements on two datasets with long-range dependencies.

**Methods And Evaluation Criteria:**

The method primarily comprises three components: differentiating the Personalized PageRank (PPR) process, employing techniques to accelerate PPR, and integrating a short-range message passing module. Extensive experiments on multiple datasets are conducted to evaluate the effectiveness of the proposed method.

**Other Comments Or Suggestions:**

No

**Other Strengths And Weaknesses:**

No

**Questions For Authors:**

No

**Relation To Broader Scientific Literature:**

It contributes to GNN.

**Theoretical Claims:**

The theory demonstrates the availability of differentiable Personalized PageRank (PPR).

---

> ### Author Rebuttal · Authors · 2025-03-27
>
> ## Oversmoothing
>
> > While the authors claim that the proposed method mitigates oversmoothing, the experimental results do not provide a detailed analysis.
>
> We appreciate this observation regarding oversmoothing.
> As established in prior work [1,2], the incorporation of Personalized PageRank (PPR) inherently addresses the oversmoothing issue in graph neural networks.
> Since this has been thoroughly demonstrated in the literature, we did not show the mitigation of oversmoothing in our manuscript but focused our experimental validation on other novel aspects of our approach.
>
> * [1] Klicpera J, et al. Predict then Propagate: Combining neural networks with personalized pagerank for classification on graphs. ICLR, 2018
> * [2] Choi J. Personalized pagerank graph attention networks[C]. ICASSP, 2022.
>
> ## Are improvements significant?
>
> > Results in some cases do not show significant improvement over existing methods, and additional experiments or clearer explanations are needed.
>
> We respectfully submit that improvements exceeding one standard deviation in accuracy can reasonably be considered significant.
> Our method achieves this standard in 9 out of 12 datasets in Table 1 and demonstrates consistent significance when compared to MPNN baselines in Table 2.
> While we understand the desire for universal improvement across all cases, we believe this may be an overly stringent expectation given the diversity of graph datasets.
>
> To better address the reviewer's concern, we would be grateful for more specific guidance regarding what 'additional experiments or clearer explanations' would be most valuable for evaluating our method's contributions.
>
> ## The memory usage
>
> > While the algorithm's complexity is analyzed, it would be beneficial to also evaluate its memory usage with other baselines.
>
> We appreciate this suggestion regarding memory analysis.
> The attention weights of ALS are computed with the same formula in standard GAT.
> The difference is that ALS will propagate information following these attention weights for multiple iterations.
> However, the memory usage maintains the same thanks to implicit differentiation, as shown in Figure 2.
> Thus, the memory usage of ALS is the same as GAT.
>
> ## Are improvements significant with long-range dependencies?
>
> > Although the proposed method is designed to capture long-range information, it does not achieve significant improvements on two datasets with long-range dependencies.
>
> This is an insightful observation.
> We agree that when compared to Graph Transformers (GT), ALS shows more modest improvements because GTs inherently possess global receptive fields that can capture both long-range and non-adjacent node dependencies.
> However, as an MPNN variant, ALS consistently outperforms other MPNN baselines by more than one standard deviation in accuracy.
> Furthermore, the integration of ALS with GT architectures yields better performance than other MPNN-GT combinations.
> These results substantiate our claim that ALS offers superior long-range information capturing capability compared to conventional MPNN approaches.

---

### Official Review · Reviewer_cwox · 2025-03-18

**Overall Recommendation:** 2

**Summary:**

This study introduces Attentive Long-Short-range (ALS) message passing, which incorporates personalized PageRank to address the over-smoothing issue in long-range message propagation. Additionally, it utilizes implicit differentiation to effectively improve the GAT computation overhead.

**Claims And Evidence:**

While the study presents experiments demonstrating the performance of ALS across various graph types, its scope is limited to evaluating effectiveness in the node classification setting. This narrow focus raises concerns about the method’s generalizability to other graph-related tasks, such as link prediction or graph classification. Additionally, while ALS is compared against strong baselines like Graph Transformers and Graph Mambas, the study lacks a thorough analysis of computational efficiency and scalability.

**Essential References Not Discussed:**

N/A

**Experimental Designs Or Analyses:**

Detailed model settings are unavailable, making it difficult to reproduce the experimental results. Key details, such as the number of layers and hidden units used for ALS, are not clearly specified. Additionally, the study omits certain long-range graph benchmarks, which limits the comprehensiveness of the evaluation. Furthermore, there is no ablation study examining the effectiveness of different optimization techniques, leaving their individual contributions unclear.

**Methods And Evaluation Criteria:**

Yes, the proposed methods and evaluation criteria are appropriate for the problem at hand. However, the study lacks an analysis of the method’s generalizability to other graph-related tasks, such as link prediction or graph classification, limiting its broader applicability.

**Other Comments Or Suggestions:**

n/a

**Other Strengths And Weaknesses:**

The proposed method seems effective and performs well empirically.
The introduction of DPPR  is an interesting idea.
The paper's structure is clear and easy to follow.

**Questions For Authors:**

In Figure 1, there is a discussion comparing single-layered ALS to multi-layered ALS. Could you clarify why stacking multiple ALS layers is helpful for the downstream performance? Any further explanation on this would be helpful.

**Relation To Broader Scientific Literature:**

Enhancing the performance of GNNs has  implications for graph-based learning across multiple domains. This work leverages attention mechanisms to construct an attentive transition matrix for message passing, improving its ability to capture intricate data dependencies effectively.

**Theoretical Claims:**

I have checked the proofs and noted that the paper includes only one theorem. However, I find the authors' claim of proposing three acceleration techniques for expediting the computation of Differentiable PPR to be unsubstantiated. The section lacks genuine innovation, as it merely employs existing optimization techniques rather than introducing novel computational advancements. A more rigorous justification or empirical demonstration of how these techniques specifically enhance Differentiable PPR would strengthen the contribution.

---

> ### Author Rebuttal · Authors · 2025-03-27
>
> ## Link prediction and graph classification
>
> > This narrow focus raises concerns about the method’s generalizability to other graph-related tasks, such as link prediction or graph classification.
>
> > The study lacks an analysis of the method’s generalizability to other graph-related tasks.
>
> We sincerely appreciate the reviewer's valuable suggestion regarding additional evaluation tasks.
> Due to time constraints during the rebuttal period, we may not be able to conduct comprehensive evaluations across different types of graph tasks.
>
> However, we would like to note that among the PPR-based methods we referenced (APPNP, GPRGNN, PPRGo, PPRGAT, and MAGNA), only MAGNA included link prediction evaluations.
> Therefore, focusing on node classification for our DPPR-based ALS method aligns with standard practice in this research area.
>
> Furthermore, all claims in our manuscript - including memory optimization, time efficiency, and effectiveness on heterophilic graphs - have been thoroughly validated.
> Since heterophily fundamentally concerns node-level relationships, node classification tasks sufficiently demonstrate our method's capabilities in this regard.
>
> ## Efficiency and scalability
>
> > the study lacks a thorough analysis of computational efficiency and scalability.
>
> We acknowledge the reviewer's concern about computational analysis.
> The attention weight computation in ALS follows the same algorithmic approach as standard GAT.
> The key distinction lies in ALS's multiple iterations of information propagation, which typically results in computation time proportional to the number of iterations compared to GAT.
> This iteration count is primarily determined by the parameter $\alpha$ and can be significantly reduced by our three acceleration techniques.
>
> Regarding scalability, our experiments demonstrate ALS's scalability because they include two large-scale OGB datasets, and while individual subgraphs in the LRGB datasets are small, their combined scale is substantial.
>
> ## The innovation of accelerating techniques
>
> > I find the authors' claim of proposing three acceleration techniques for expediting the computation of DPPR to be unsubstantiated.
>
> We appreciate this opportunity to clarify our technical contributions.
>
> As mentioned in our discussion of AdaTerm, traditional PPR applications typically solve a single linear system, whereas we must solve H × C independent linear systems due to the multi-dimensional nature of node representations in GNNs.
> This novel challenge motivated our AdaTerm.
>
> Moreover, the multi-dimensional node representations make Krylov subspace methods difficult to apply.                                                                                                                           While Krylov methods may require dozens of basis vectors to construct subspaces (negligible overhead for single systems), combining them with GNNs would require independent subspaces for each representation dimension, leading to H × C times of memory usage.
> This inspired our SymGAT, which enables more memory-efficient conjugate gradient algorithms.
>
> Regarding EigenInit, since PPR was originally designed for homophilic graphs like page recommendations where EigenInit showed limited benefits, traditional PPR methods did not investigate into it.
> However, our experiments demonstrate that EigenInit provides significant acceleration on heterophilic graphs - an important finding given the growing interest in heterophilic graph research.
>
> In summary, the three techniques were specifically designed to enhance PPR-based GNN methods, including our DPPR approach.
>
> ## Experimental Designs Or Analyses
>
> We respectfully disagree with this particular critique.
>
> We have provided detailed experimental settings in Appendix E and included reproduction scripts with submitted code.
> We omit other LRGB datasets because they are inadequate for the evaluation, as explained in the footnote on page 6.
> Furthermore, Appendices C and D contain extensive ablation studies - we believe these address the reviewer's concerns about 'different optimization techniques'.
> We would welcome further clarification if we have misunderstood the reviewer's point.
>
> ## Multi-layered ALS
>
> > why stacking multiple ALS layers is helpful?
>
> This is an excellent and commonly raised question about many methods.
>
> As shown in Appendix C of the IGNN paper, stacking multiple layers is theoretically equivalent to using a single wider layer.
> However, empirical results demonstrate that multi-layer IGNN architectures achieve substantially better performance on datasets like PPI.
> A similar phenomenon occurs with Graph Transformers (GT) - while a single GT layer can theoretically access all global information, deeper architectures typically perform better.
> Our work validates this design pattern's effectiveness, though the underlying reasons remain an open research question.
>
> We hope this common architectural consideration won't negatively impact the evaluation of our contributions.

---

### Decision · Program_Chairs · 2025-05-01

**Decision:**

Reject

**Comment:**

The reviewers unanimously recommend rejecting the paper and I agree with their recommendation. Several reviewers noted a disconnect of the initially presented motivation of the work stemming from the mitigation of the oversmoothing issue and the subsequent solution approach and experiments. I agree with the reviewers, that references to the literature are insufficient in this context and further theoretical or empirical work should be done to evidence the advantages of the proposed method when considering the oversmoothing problem. The reviewers furthermore noted that the paper's clarity and presentation should be improved. While the rebuttal of the authors addressed several of the concerns raised by the reviewers. I agree with the reviewers, that the submitted manuscript is not ready for publication at this stage. I hope that the reviewer's comments will help the authors improve their manuscript.